# Name Disambiguation Scheme Based on Heterogeneous Academic Sites

**Dojin Choi** [1], **Junhyeok Jang** [2], **Sangho Song** [2], **Hyeonbyeong Lee** [2], **Jongtae Lim** [2], **Kyoungsoo Bok** [3] **and Jaesoo Yoo** [2,*]

1. Department of Computer Engineering, Changwon National University, Changwondaehak-ro 20, Uichang-gu, Changwon-si 51140, Gyeongsangnam-do, Republic of Korea; dojinchoi@changwon.ac.kr
2. Department of Information and Communication Engineering, Chungbuk National University, Chung-dae-ro 1, Seowon-gu, Cheongju 28644, Chungcheongbuk-do, Republic of Korea; tkalfm8@naver.com (J.J.); ssh@cbnu.ac.kr (S.S.); lhb@cbnu.ac.kr (H.L.); jtlim@cbnu.ac.kr (J.L.)
3. Department of Artificial Intelligence Convergence, Wonkwang University, Iksandae 460, Iksan 54538, Jeollabuk-do, Republic of Korea; ksbok@wku.ac.kr
* Correspondence: yjs@cbnu.ac.kr; Tel.: +82-43-261-3230

**Abstract:** Academic researchers publish their work in various formats, such as papers, patents, and research reports, on different academic sites. When searching for a particular researcher's work, it can be challenging to pinpoint the right individual, especially when there are multiple researchers with the same name. In order to handle this issue, we propose a name disambiguation scheme for researchers with the same name based on heterogeneous academic sites. The proposed scheme collects and integrates research results from these varied academic sites, focusing on attributes crucial for disambiguation. It then employs clustering techniques to identify individuals who share the same name. Additionally, we implement the proposed rule-based algorithm name disambiguation method and the existing deep learning-based identification method. This approach allows for the selection of the most accurate disambiguation scheme, taking into account the metadata available in the academic sites, using a multi-classifier approach. We consider various researchers' achievements and metadata of articles registered in various academic search sites. The proposed scheme showed an exceptionally high F1-measure value of 0.99. In this paper, we propose a multi-classifier that executes the most appropriate disambiguation scheme depending on the inputted metadata. The proposed multi-classifier shows the high F1-measure value of 0.67.

**Keywords:** name disambiguation; author name disambiguation; deep learning; multi-classifier; HAC





## 1. Introduction

Generally, users enter specific keywords on academic search sites to find items through scholarly databases. These sites provide scholarly data, such as articles and reports, that match these keywords as search results. These search results include articles that provide information about the authors and contents of the articles. Since academic search sites hold several research records, individuals with the same name, even in the same research field, are commonly encountered. That is, researchers studying in the same or different fields often have identical names. Most academic search sites offer a feature to research within the author name search results. This feature is provided to refine and search again for the researcher or keyword the user actually wants to find. However, this feature poses a challenge, as users are required to determine for themselves from the search results if the name belongs to a different researcher. Additionally, even if a specific academic search site distinguishes between individuals with the same name effectively, determining those based on results provided by different academic search sites is very challenging. This is inconvenient and is a basis for incorrect judgments made by individuals searching for academic information. Therefore, in order to utilize various academic search sites, a

function that can identify individuals with the same name across these different academic search sites is necessary [1,2].

Distinguishing and identifying individuals with the same name play a significant role in enhancing search accuracy. When users find the name of a researcher, the search results provide all the research outputs of every researcher with the same name. By using the filtering feature provided by academic search sites and entering additional information about the desired content, users can increase the accuracy of the search results. Studies on name disambiguation have been conducted using schemes that use the metadata of academic search sites to identify authors with identical surname and given name [1–22]. Various studies have been conducted to address name disambiguation on academic sites. In [5], a scheme was proposed to discern individuals with the same name using the metadata of academic search sites and determining name matches based on the similarity of attributes between two different papers. Furthermore, studies have been conducted to establish rules for calculating the similarity of attributes between two different papers and conduct cluster analysis based on the calculated similarity [6,7]. A scheme that uses the metadata of a paper as a feature of deep neural networks has been proposed to discern individuals with the same name [10–15]. A study in which individuals with the same name could be discerned by modeling a graph-based on the attributes of papers and author information and by using a graph auto-encoder was also conducted [17]. Some name disambiguation schemes have been studied [18–20]. Some of these proposed a taxonomy on the name disambiguation schemes, such as supervised learning, unsupervised learning, graph-based, semi-supervised, and heuristic-based, and also explored evidence- based schemes. They described the characteristics of each scheme, including its performance, strengths, and limitations. Others proposed initial-based methods based on bibliographic datasets in which the true identities of authors are known [21]. They insist that the first initial-based method already correctly identifies 97% of authors [22] and proposed an author name disambiguation framework by using knowledge graph embedding. The framework extracts entity features from a scholarly knowledge graph (SKG) to represent dense representation as low-dimensional vector space to perform HAC. Recently, various studies have used graph neural networks and graph embedding to perform learning based on graph modeling of papers and author information to discern individuals with the same name [11–15]. However, these existing schemes only use structured datasets. In actual academic search sites, the available metadata vary across sites; thus, research that considers this should be conducted. For instance, in academic search sites where specific metadata do not exist, weight learning for such metadata cannot be performed, necessitating research to address this gap. In addition, since users may be seeking research materials published on different sites, searching and collecting data from all these sites are essential. Lastly, a method that analyzes name disambiguation based on the information collected from different academic search sites is needed. Information collected from two or more sites may contain overlapping papers along with those that exist on only one site. Name disambiguation is imperative in such an environment.

In this paper, we propose a name disambiguation system that enables name disambiguation analysis across different academic search sites by collecting papers from currently active academic search services. The proposed scheme conducts rule-based name disambiguation analysis that can operate dynamically based on metadata from different academic search sites. In addition, we propose a multi-classifier according to the characteristics of metadata. The proposed classifier selects a more accurate scheme between the existing name disambiguation schemes and the proposed scheme based on metadata characteristics. The proposed multi-classifier provides a feature to flexibly perform name disambiguation based on the input metadata. The excellence and validity of the proposed method are demonstrated through various performance evaluations.

This paper is organized as follows. The term "Name Disambiguation" is defined, and the characteristics and problems of existing name disambiguation schemes are described in Section 2. The proposed name disambiguation method is detailed in Section 3. The

better fitting of the proposed method is demonstrated through a comparative analysis with existing schemes in Section 4, and the study is concluded, along with future research directions, in Section 5.

## 2. Related Work

### 2.1. Name Disambiguation

Rule-based name disambiguation schemes extract distinguishable attributes of authors from papers and create rules using these attributes. Each rule incorporates a weight, which is determined dynamically according to performance evaluations, and these weights are applied in cluster analysis.

In [7], a rule-based name disambiguation scheme was proposed. Figure 1 shows the flow chart of the rule-based name disambiguation scheme. When candidates with the same name input, a document vector is created based on the articles written by the candidates. Then, the rule-based heuristic similarity scores are calculated. Finally, they perform a clustering method, such as HAC, based on the similarity scores. Finally, they obtain dis-ambiguous data by generating clusters with the same authors. The collected data were preprocessed based on established rules. The preprocessed data were then subjected to name disambiguation using one of the two schemes: the rule-based name disambiguation method or a classifier-based method. After collecting documents from the database, attributes, such as surname, first name, co-authors, affiliation, research field, and keywords, were extracted for the name disambiguation scheme. The surname extracted during the preprocessing stage was used as is, while the initial of the first name was included in a data block, along with the attributes. Rule-based similarity was calculated on a block-by-block basis, and hierarchical agglomerative clustering (HAC) was performed based on the calculated similarity. HAC merges the most similar clusters together, starting with each data point as a separate cluster. HAC is most suitable in the author name disambiguation studies. For example, k-means clustering requires the number of cluster k. This is not suitable because the exact number of authors with the same name is unknown. The "similarity estimated by classifiers" method performed clustering based on similarity scores generated by classifiers and extracted stems from paper titles, abstracts, and keywords. For the extracted stems, similarity was calculated using the term frequency-inverse document frequency (TF-IDF) and latent semantic analysis models. The classifiers were trained using the information from data blocks. HAC was performed based on the similarity scores generated by the classifier. Name disambiguation was performed based on the results of the HAC.

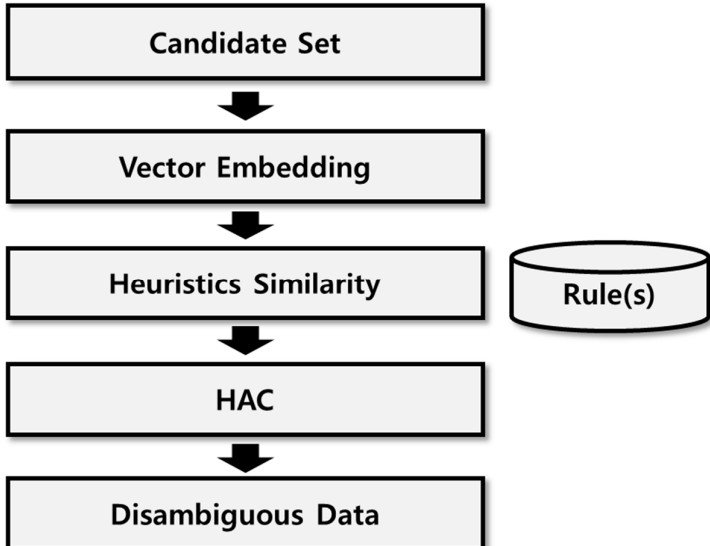

**Figure 1.** A flow chart of a rule-based name disambiguation scheme.

A deep learning-based name disambiguation schemes generate weights using document attributes, converts these into vector values, and then proceeds with deep learning. The values obtained through learning are converted into inter-document distance values, after which cluster analysis is conducted to disambiguate names. The weights for deep learning represent document attributes in the form of graph data, extracting distinguishable attributes among multiple documents to create adjacency and feature matrices. This information is applied to a graph convolutional neural network (GCN) [23] to learn feature vectors. Name disambiguation is then performed based on the learned feature vectors using HAC. Figure 2 shows a flow chart of a deep learning-based name disambiguation scheme. Candidates with the same name enter as in the rule-based scheme. Then, it constructs graphs based on the papers written by the candidates. The constructed graphs are used in GCN. The papers of the candidates are also used to make feature vectors to represent the vertex features of the graphs. It performs GCN, which is a graph-based deep learning model, on the graphs by using feature vectors. Finally, the learned feature vectors are utilized to perform HAC. The papers written by the identical author are grouped as one cluster.

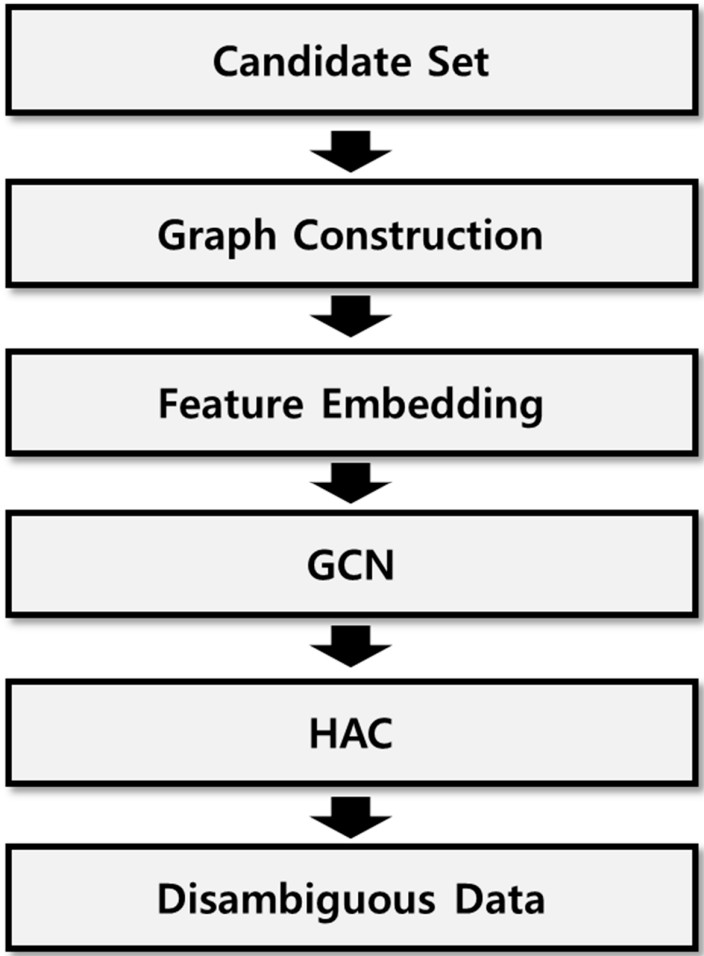

**Figure 2.** A flow chart of a deep learning-based name disambiguation scheme.

A Graph Convolutional Network (GCN) is a specialized type of graph neural network designed to comprehend the connections between vertices in a graph data structure for tasks such as predicting associations or classifying vertices. It introduces a convolution layer to more effectively improve the learning of attribute vectors compared to traditional graph neural networks. In GCNs, the graph data consist of vertices and edges, and computations involve weight sharing. The same filter is used to train all vertices in the graph in the weight sharing. During the weight sharing process, redundant attributes operate with the

same weights, which enhances the correlation between nodes that have edge relationships. By updating the information for all vertices in this manner, vector values can be determined for cluster analysis.

A name disambiguation scheme based on a GCN was proposed previously [12–14]. From all documents, those with name ambiguity were selected, and name-specific sets were formed to create candidate groups. During the global representation learning process, the attribute information (titles, keywords, co-authors, affiliations, and conferences) for all documents in the selected name-specific sets was extracted to form attribute data. The attribute information was segmented into individual words and then converted into vector values using Word2Vec. A feature matrix of the document, based on attributes, was created following the TF-IDF process using the transformed vector values. In the "three association graphs" process, edge creation conditions were set for name-specific sets of the candidate group. If the conditions exceeded a threshold, an edge relationship formed between two vertices, resulting in the creation of an adjacency matrix that integrated the edge information between vertices. The types of graphs produced included paper-to-paper graphs, co-author graphs, and paper-to-author graphs. The GCN was performed using the created adjacency and feature matrices. Ultimately, based on the learned feature vectors, HAC was carried out for name disambiguation.

### 2.2. Limitations of Previous Studies

Traditional name disambiguation schemes use pre-constructed structured datasets. Name disambiguation on actual research materials from heterogeneous academic search services in operation faces an issue: if the presence or absence of metadata in the name disambiguation is not considered, direct application of the name disambiguation scheme becomes unfeasible. Furthermore, as academic search services vary in the type of research materials they offer based on their purpose, different academic services must be searched to review the works of a researcher published in various formats. Even if the research materials are of the same type, the absence of shared metadata between different academic search services can make name disambiguation exceedingly difficult without separate preprocessing. For instance, while some research materials may list affiliations as general as "Chungbuk National University", others might provide detailed affiliations such as "Chungbuk National University, Information and Communication Engineering", necessitating data preprocessing.

Additionally, as academic search services such as Scopus, Web of Science, and google scholar provide research materials specific to their own purposes, finding all the works of an author in one academic search site can be challenging if they have published different types of research materials. For example, if an author publishes a paper based on a particular research project and produces reports or patents as research outcomes, searching for these different types of research materials within a single academic search service becomes very difficult. Ultimately, users have to search for research materials on multiple academic search services. Hence, in this study, we collected research materials from various operational academic search services and performed name disambiguation analysis on the collected material. We also considered the metadata from various academic search services to apply a uniform preprocessing approach. This led to the advantage of identifying potential attributes to consider in the name disambiguation scheme. The main motivations and significances of this paper are as follows:

1.  Academic search sites are diverse. There is a need for a name disambiguation scheme that considers various metadata of papers registered in various academic search sites.
2.  Researchers' achievements are diverse such as papers, conference papers, technical reports, projects, and patents. Therefore, we need a name disambiguation scheme that considers various researcher's achievements.
3.  Some name disambiguation schemes are only suitable for specific metadata. It is difficult to create a name disambiguation scheme that is suitable for all languages or metadata. Therefore, it is necessary to provide a name disambiguation scheme that is suitable

for the inputted metadata. In this paper, we propose a multi-classifier that executes a different scheme to disambiguate names depending on the inputted metadata.

## 3. Proposed Name Disambiguation Scheme

### 3.1. Overall Structure

Traditional name disambiguation schemes, such as rule-based [7] and deep learning-based schemes [13], use structured datasets. In this study, a method for name disambiguation by directly collecting data from multiple operational academic search sites is proposed. The proposed method standardizes affiliation information using a dedicated affiliation table when the affiliation is a university. Then, rules are defined using commonly used metadata to disambiguate names.

The metadata provided by existing academic search sites is diverse. Name disambiguation schemes do not consider the diversity of metadata provided by each academic search site. To consider this diversity of metadata, a method should selectively execute the name disambiguation scheme based on the input data and exhibit the best performance. Additionally, even if some metadata can be used in the proposed method, if the expected performance is inferior to that of existing schemes, applying the existing methods is more effective. We propose a multi-classifier scheme that considers all these situations. When metadata are input, the multi-classifier is performed based on the rule-based scheme proposed in this paper, as well as the existing deep learning-based name disambiguation method.

The multi-classifier uses limited metadata from the actual data required by each method to select the most suitable scheme for name disambiguation. The multi-classifier is designed in an expandable manner to consider also new name disambiguation schemes that may emerge in the future.

Figure 3 presents the overall system architecture of the proposed scheme. The collector gathers research papers, project data, and affiliation information for research outcomes from academic search services. The preprocessor creates a set of name-ambiguity candidates with identical names, considered as subjects in this study, from all the documents collected by the collector. The collected affiliation information is transformed into a standardized form using an affiliation table. Attributes to be used for name disambiguation are then extracted. In the analyzer, the preprocessed attribute data are used to analyze the similarity between all documents of the name-ambiguity candidate group using both the rule-based and deep learning schemes. The analyzer performs calculation simultaneously for the rule-based candidate group similarity and the deep learning-based candidate group similarity. Finally, in the discriminator, the analyzed document similarity data are represented as a distance matrix by using sim2diss function for clustering execution, and HAC is performed to divide the clusters by unique author documents and to disambiguate names.

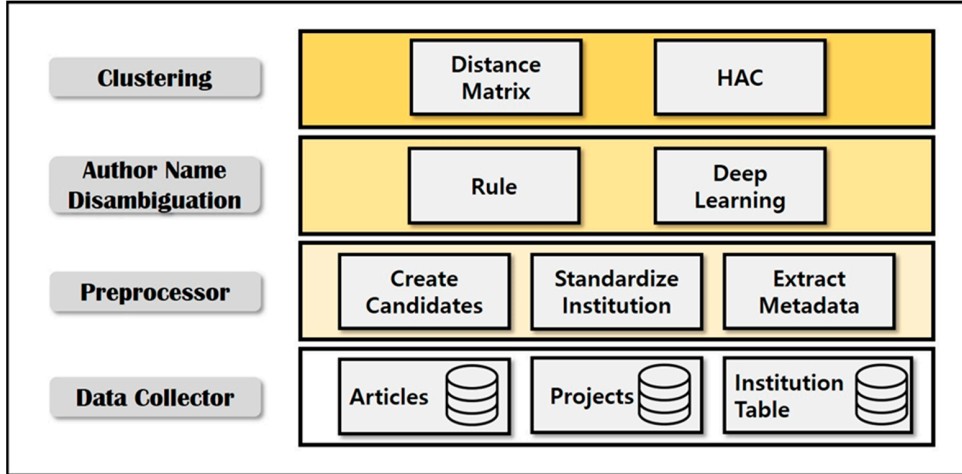

**Figure 3.** System architecture of the proposed scheme.

*3.2. Data Collector*

This study aimed to collect documents from heterogeneous academic search services, including research papers, as well as national R and D, patent, and research reports. Since heterogeneous forms of documents were collected from heterogeneous sites, understanding the metadata necessary for name disambiguation was crucial.

The collector, upon user keyword input, collects the attribute information of documents appearing in the keyword search results and stores it in the internal database. The collected data are used as attributes for analysis in the name disambiguation analyzer after preprocessing in the next preprocessing step. When collecting the material, understanding the metadata used by the heterogeneous academic search services providing research outcomes is essential. For instance, academic search services, primarily offering papers, contain metadata distinguishing between academic journals and conferences. However, sites that primarily offer project information do not have such distinguishing metadata. Among them, whether they support metadata to identify international journal listings also varies. Moreover, domestic academic search sites that use journal information as metadata express it in various forms, such as journals, academic journals, and proceedings, which should be considered during data collection. We use type information about an article that the academic document cites, such as ScienceOn, KCI, NTIS and DBpia provide, to distinguish journals, academic journals, and proceedings.

Next, understanding the attributes of various types of documents is essential. In academic search services, a significant proportion of authors publishing papers are affiliated with universities. However, for sites providing R and D information, authors publishing research outputs have various affiliations, including research institutes, national departments, companies, and universities. Furthermore, the authors of papers can be categorized into the main, co-, and corresponding authors. However, for R and D, the authors consist of participating researchers and research leaders. Thus, depending on the type of research output, the nature and form of attributes differ. Understanding the meaning of similar attributes and collecting them accordingly are crucial steps.

The collector gathers all data usable for name disambiguation schemes. The research outputs of an author stored in the database after collecting the necessary values from academic search sites are listed in Table 1. All metadata that can be collected from academic search services are gathered. Among the collected metadata, the commonly used attributes and attributes for which advantageous weights can be given for name disambiguation are identified. Commonly usable metadata include the author name and affiliation, co-authors, title of the document, document keywords, and publication year. The attributes that can be given favorable weights for name disambiguation include the research field, email, academic journals, and academic conferences.

**Table 1.** Example of collected attributes.

| Feature | Paper 1 | Paper 2 |
|---|---|---|
| title | An Author Name Disambiguation Method Considering Metadata Features | Development of Fuel Cell System Considering Weight |
| abstract | A same-name identification scheme that considers metadata to identify people with the same name on heterogeneous sites... | Energy commercialization considering weight using an ultra-light tube-type fuel cell system... |
| Keywords | Name disambiguation, Metadata | Fuel cell, Tube type |
| Year | 2018 | 2020 |
| Affiliation | Chungbuk National University | Pohang University of Science and Technology |
| First Author | Junhyeok Jang | Junhyeok Jang |
| Co-author | Sanghyeok Kim, Yuna Kim, Dojin Choi, Jaesoo Yoo | Taehyeong Kim, Jinyong Lee, Sunkyu Han, Minkyo Lim |
| Journal | Big data technology journal | Resource technology journal |
| Publisher | Big Data Society | Society for New and Renewable Energy |
| e-mail(s) | dataman@kakao.com | azeez448@nate.com |
| Research area | bigdata | Energy, resource tech |
| Research Period | 2018~2020 | 2018~2022 |

The attributes of research outputs used in the proposed scheme are listed in Table 1. Since the proposed method considered various academic search services, it used commonly existing metadata, such as the author name and affiliation, co-authors, publication year, academic journals, and academic conferences, as attribute values for rule-based name disambiguation schemes.

### 3.3. Preprocessing

Data collected directly from the collector cannot be used as they are; therefore, preprocessing is required. In the preprocessor, all documents containing the same author name are gathered as potential name disambiguation candidates. Attributes needed for name disambiguation analysis are then extracted from the documents. At this point, affiliation information is normalized using the affiliation table.

From the collected documents, candidate groups for name disambiguation need to be generated to narrow down the set of documents that can be considered as potential name matches. Name disambiguation is performed using document similarity within the created candidate sets. In this study, two or more documents with the same author name are considered as candidate groups.

Table 2 shows an example of a name disambiguation candidate group. Collected data containing two or more documents with the author name "Jang Jun-hyeok" were generated as name disambiguation candidates. Within the name disambiguation candidate group, attributes, such as the title of the paper, affiliation, publication year, co-authors, journals, and academic conferences, were collected. To help explain the name disambiguation scheme proposed in this paper, documents "Jang Jun-hyeok_0", "Jang Jun-hyeok_1", and "Jang Jun-hyeok_2" represented unique research outputs of a single Jang Jun-hyeok author, while document "Jang Jun-hyeok_3" represented a research output of a different Jang Jun-hyeok with the same name. The proposed scheme constitutes the name disambiguation document candidate group from all research outputs with the same author name, regardless of the type of authorship (main author, co-author, or corresponding author). Each document in the name disambiguation candidate group was labeled with the author name, and numbers were appended after the name to distinguish documents.

**Table 2.** Example of name disambiguation candidates.

| Name | Title | Inst. | Year | Co-Author | Journal |
|------|-------|-------|------|-----------|---------|
| Jang Jun-hyeok_0 | Author Name Disambiguation Tasks Considering Metadata. | Chungbuk National Univ. | 2018 | (S. H. Kim, Y. A., Kim, D. J. Choi, J. S. Yoo) | Bigdata Society |
| Jang Jun-hyeok_1 | Metadata Learning by using Machine learning. | Chungbuk National Univ. | 2021 | (Y. A. Kim, D. J. Choi, J. S. Yoo] | Bigdata Society |
| Jang Jun-hyeok_2 | Pitcher's Contribution to ERA. | Sports Science Tech. | 2022 | (D.J. Choi, J.S. Yoo) | Bigdata Society |
| Jang Jun-hyeok_3 | Fuel Cell system considering. | Pohang Univ. | 2020 | (T. H. Kim, J. Y. Lee, S. K. Han, M. G. Lim) | New and Renewable Energy. |

In this study, we normalized the affiliations listed in documents. The listing of affiliations (when the author affiliation is a university) can vary across academic search services. Furthermore, authors may have different styles of listing their affiliations. For example, an affiliation Pennsylvania State University, depending on different styles, can be written as Pennsylvania State University, Pennsylvania State Univ., PSU, Penn. State Univ., or Penn. State College, among other variations. Additionally, in some instances, such as "Information Sciences and Technology, Penn. State Univ.", a specific department or detailed affiliation information is included. For such cases, a standardized affiliation form needs to be normalized. Web of Science, a globally renowned academic search service, provides affiliation metadata to alleviate confusion caused by various affiliation entries and to verify various forms of affiliation information. Leveraging this, the same affiliation information

can be normalized. In our research, affiliations listed in various forms were normalized to a unified format. All institution names and their synonyms listed on academic search services were stored in a database. Based on the stored information, synonymous institution names were standardized into a representative institution name for affiliation notation. Affiliations with detailed information (e.g., departments) were converted into only the university name to efficiently process the affiliation information.

After creating the name disambiguation candidate group, attributes to be used in the name disambiguation algorithm were extracted. Notably, the variation in metadata across academic search services must be considered. Some academic search services provide information only about the paper, while others offer data like R and D research reports. The diverse metadata provided by academic search services must be preprocessed into a commonly usable format. The preprocessor extracted and used only the metadata to be inputted into the name disambiguation analyzer.

### 3.4. Author Name Disambiguation

The author name disambiguation scheme uses the candidate group created by the preprocessor. It calculates the similarity between documents within the candidate group to compare whether two documents were written by identical authors. The name disambiguation scheme defines a method to calculate the similarity between attributes of two documents. It computes the sum of similarities between attributes to determine the final similarity score. To discern significant attributes, weights are assigned to attributes based on both the rule-based and deep learning schemes.

#### 3.4.1. Rule-Based Scheme

The attributes to be used in the name disambiguation analyzer were extracted by the preprocessor. The name disambiguation analyzer defined rules for disambiguating names and assigned weights based on the importance of each rule. The similarity calculation rules and weights of the proposed rule-based scheme are listed in Table 3. The proposed rule-based approach first performed name disambiguation based on exception cases. An exception case refers to a specific rule that determines whether two documents are written by the same author. If a document does not fall under an exception case, the similarity is calculated according to the proposed four rules. The similarity values for each rule are summed up, and if the total similarity exceeds a certain threshold, the research is determined to have been authored by the same individual.

**Table 3.** Attribute and weight application rules.

| Name | Rule | Weight |
|---|---|---|
| Exception Case | The titles are exactly the same<br>The author affiliations are exactly the same<br>The co-authors are exactly the same | 4 |
| Affiliation | Jaro–Winkler Similarity [24]<br>$s_j = \begin{cases} 0, & m = 0 \\ \frac{1}{3}\left(\frac{m}{\|s_1\|} + \frac{m}{\|s_2\|} + \frac{m-t}{m}\right), & otherwise \end{cases}$<br>$s_w = s_j + \omega p(1 - s_j)$ | 0~1 |
| Year | Difference of publication year<br>$p = -\left(\frac{\|yd_1 - yd_2\|}{cy}\right) - 1$ | 0~1 |
| Co-author | Number of identical co-authors<br>$c = \frac{1 - e^{-\|x\|}}{2}$ | 0~0.5 |
| | Proportion of identical of co-authors<br>$r = \frac{x/y}{2}$ | 0~0.5 |
| Venue | Exactly the same or not | 0 or 1 |

First, if the two documents match the exception case attributes, they are awarded four points, and other attributes are not considered. The first exception case is when the titles of the two documents being compared are the same. The second exception case is when, after normalizing the affiliation of the document through the affiliation table in the preprocessor, the affiliations are found to be identical. In case of a discrepancy in the detailed affiliation, the Jaro–Winkler [24] similarity, a method that considers the number and position of common characters between two strings, was used to calculate the similarity of affiliations. If the Jaro–Winkler similarity score exceeded a predefined threshold, the documents were deemed to have the same affiliation.

The publication year attribute represents the difference in publication years between the two papers. The year rule represents the weight calculation based on the publication year (p). Here, $yd_1$ and $yd_2$ indicate the publication years of the two documents, and $cy$ represents the publication year span set by the user. In this study, the volume of documents collected by the collector varied widely depending on the keywords entered when collecting documents. Moreover, when new subject keywords emerge, past data need not be collected. Therefore, setting the publication year when collecting documents was necessary.

For the co-author count attribute, the number of identical co-authors in the two papers was compared. The rule of the number of identical co-authors displays the weight calculation based on the number of co-authors (c), where x is the number of identical co-authors between the two documents. The co-author ratio attribute represents the ratio of identical co-authors to the total number of co-authors in the two papers. The rule of the proportion of identical co-authors - represents the weight calculation based on the co-author ratio (r). In this context, $x$ is the number of identical co-authors, similar to the co-author count formula, and $y$ is the total number of co-authors in the document with more co-authors among the two being compared.

Regarding the journal and conference attributes, a comparison was made between the publication venues of the two papers. If the two documents were identical venues, a weight of one was assigned. If the venues were different or did not include the journal and conference attributes, a weight of zero was allocated.

Figure 4 shows an example of weight determination based on the rules listed in Table 3. For the affiliation attribute, the Jaro–Winkler distance value was derived from the redundant words "Chungbuk National Univ." and "Chungbuk National Univ. Bigdata Depart", resulting in the $s_j$ value. In the Jaro–Winkler similarity, $m$ denotes the number of common characters in the two strings. In the above example, all the letters of $s_1$ are common. m is calculated as 21. Therefore, the $s_j$ value is 0.7. To calculate $s_w$, the common prefix length $w$ is set to 4 (maximum value), and $p$ is calculated as a standard value of 0.1. As a result, $s_w$ is 0.83. The publication year attribute defines the data collection period as 5 years. By using the absolute value function to calculate the publication year difference between the two documents, a score of 0.4 was assigned. The co-author attribute determines the number of identical co-authors in the two papers, excluding the co-author with the same name, "Jang Jun-hyuk". The co-author ratio divides the number of co-authors by the number of co-authors in the document with the larger number of co-authors. If we evaluate only the number of co-authors in a paper written by one author, we get an ambiguous value of approximately 0.63. This value is a very uncertain number to perform an analysis with the same name. In addition, if the number of co-authors of the same name accidentally increases in a paper with a large number of authors, the value of the number of co-authors converges to 1. To solve this problem, we supplemented this by adding a co-authors ratio. Here, the combined values of the co-author count and ratio were halved, and scores of 0.475 and 0.375, respectively, were obtained. Thus, the co-author attribute value was represented as 0.85. For the journal and conference attributes, since conferences of both the documents were "Journal of Bigdata", they were identical, and a score of one was assigned. By adding the results from all the rules, the final weight was determined. The weight of the sample documents "Jang_0" and "Jang_1" was 3.08, indicating a high similarity.

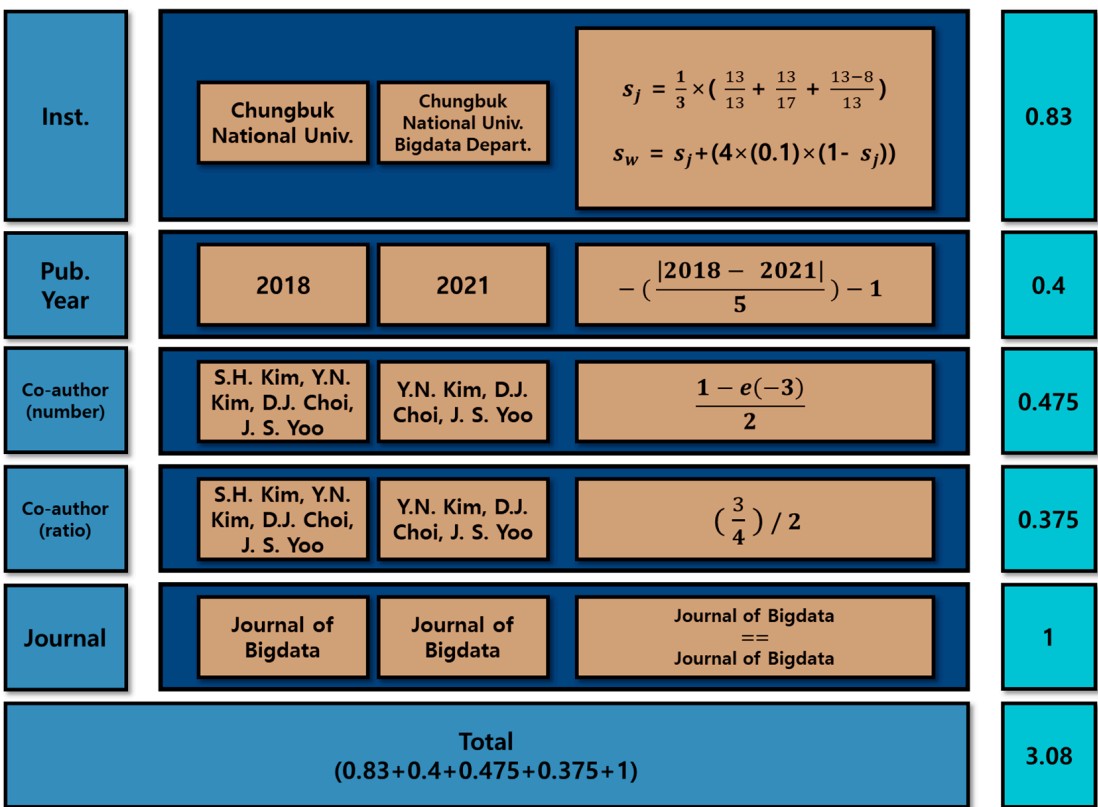

**Figure 4.** Example of name disambiguation rules.

3.4.2. Deep Learning-Based Scheme

The deep learning-based analysis, similar to rule-based analysis schemes, employs major attributes from the metadata that can act as distinguishing factors for individuals with the same name. These attributes were represented in a graph, and the GCN was utilized to learn the latent features of the paper.

Deep learning analysis uses various attributes, such as the title, keywords, abstract, co-authors, publication year, and journal data. As the document title and abstract were used as attributes, major keywords from these attributes were extracted using natural language processing packages, such as konlpy and NLTK. The extracted keywords were then converted into a vector form using FastText. This converted vector was used as the input vector for the deep learning model. In this study, the converted vectors were constructed in the form of a triplet network. The triplet network structure, as shown in Figure 5, represents vectors by placing vectors with similar and dissimilar values close to each other and further apart, respectively. In the (A) triplet network, anchor refers to a document of a unique author, positive refers to another document of that unique author, and negative refers to a document of an author with the same name but not the unique author. With the anchor document as a reference, the objective of the triplet network is to bring the documents corresponding to the positive closer and drive those corresponding to the negative far apart. As shown in Figure 5b, a pid triplet transformed the vector value of the converted paper into an optimal vector value using the vector value of another paper. Initially, $P_{pj}$ referred to a paper similar to the pid paper (with author, title, publication year, keywords, journal info, etc.), and $N_{pk}$ referred to a paper dissimilar to the pid paper. The aim of the triplet was to calculate the distance between vectors, bringing similar vectors closer and pushing dissimilar vectors further apart. Therefore, based on the similarity in the information of the papers, the vector values of all the input papers were converted into the form of a triplet network, as shown in Figure 5a.

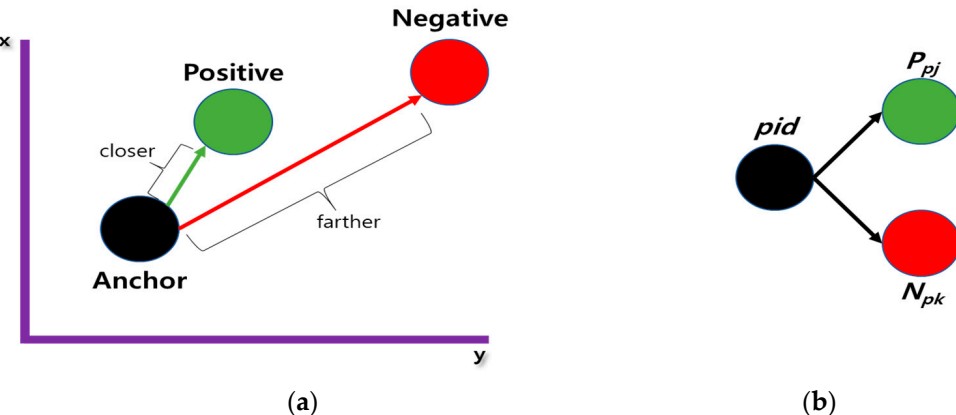

(a)                    (b)

**Figure 5.** Triplet network. (**a**) Triplet network, (**b**) Pid triplet.

*3.5. Clustering*

To distinguish between individuals with the same name, documents in the pool of potential matches must be clustered by unique authors. In the clustering stage, the similarity values generated from the previous name disambiguation analyzer were used. The similarity values were converted into distance values to be used as input data for cluster analysis.

In this section, discerning between individuals with the same name using both rule-based and deep learning schemes of the name disambiguation analyzer is discussed. Using the similarity between documents of authors with the same name, determining whether the author of a paper being compared is indeed the same author is necessary. First, to convert inter-document similarity into a distance value used in HAC, a distance matrix transformation was performed. Using the converted distance values, HAC was executed to distinguish between individuals with the same name.

HAC used in the name disambiguation scheme compared distances between clusters to perform clustering. For this, the document similarity generated by the name disambiguation analyzer was converted into a distance value. The inter-document similarity was represented in the form of a similarity matrix. The process of converting the similarity matrix into the distance matrix using the distance conversion formula, sim2diss, is explained next.

The similarity matrix is symmetrical in nature; hence, N (N − 1)/2 pairs of a,b were generated. The distance value was computed for all pairs, and the results were organized in matrix form. Since distance values were calculated for every a,b pair, the matrix was a square matrix. The similarity values ranged between zero and four, based on a weighted application rule with four as the maximum score. The diagonal terms of the matrix were "0", as they represented the distance to oneself, making the matrix symmetrical about its diagonal. Regarding the size of the similarity matrix: in case of $d$ documents, it compared all documents of authors with the same name to generate inter-document similarities, representing them in a $d \times d$ matrix.

Figure 6 illustrates the method of constructing the similarity matrix using inter-document similarities. The input data A, B, and C have a total of three distance values, and the distances between all pairs, such as A–B and A–C, can be represented as a $3 \times 3$ matrix. Figure 5a displays a graph containing inter-document similarity values calculated from the name disambiguation analyzer. The distance between input data A and B is represented as one, A and C as three, and B and C as two. Figure 6 presents the similarity matrix in matrix form and shows all distance values from the graph data.

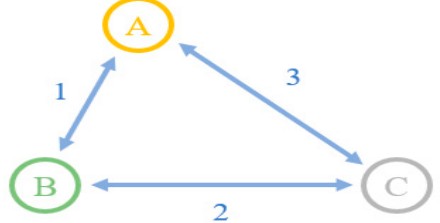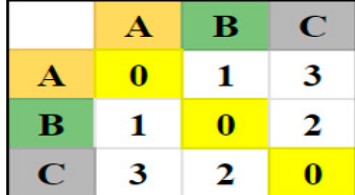

**Figure 6.** Example of the creation of a similarity matrix.

An example of representing inter-document similarity values in the form of a similarity matrix is listed in Table 4. Each row and column, such as Jang_0, Jang_1, Jang_2, and Jang_3, represents candidate documents with the same name. As every document is compared 1:1 for similarity, it forms a symmetrical matrix. Each element in the symmetrical matrix signifies the similarity values between the compared documents. Documents Jang_0 and Jang_1 exhibited a similarity value of four; therefore, both the documents were concluded to have been written by the same author.

**Table 4.** Similarity matrix example.

|        | Jang_0 | Jang_1 | Jang_2 | Jang_3 |
|--------|--------|--------|--------|--------|
| Jang_0 | 0      | 4      | 1.9    | 0.6    |
| Jang_1 | 4      | 0      | 2.6    | 0.8    |
| Jang_2 | 1.9    | 2.6    | 0      | 0.6    |
| Jang_3 | 0.6    | 0.8    | 0.6    | 0      |

HAC calculates the distance between clusters and performs clustering based on these distance values. Therefore, in this study, we converted similarity values into inter-cluster distances. Equation (1) is a function provided in the statistical solution program R, which converts inter-document similarity into the inter-document distance value, known as the distance matrix, using the sim2diss formula.

$$1 - \left(\frac{Similarity}{MAX}\right) \tag{1}$$

The rule-based name disambiguation scheme represents the generated similarity as a distance value using rules based on document attribute values. Therefore, *MAX* in Equation (1) corresponds to the total number of attributes. When the similarity of two documents is indicated as a perfect score of four points, the distance value of these two documents can be expressed as $1 - (4/4) = 0$. Since it represents the distance between two documents, a higher similarity results in a smaller distance value. The range of the distance value is between zero and one. Contrary to when calculating similarity, a distance value closer to zero indicates higher similarity between the two documents.

The similarity matrix example from Table 4, which utilizes the sim2diss formula, was represented in the form of a distance matrix, as summarized in Table 5. After calculating the sim2diss formula for all similarities, it was represented in matrix form. Documents Jang Jun-hyuk_0 and Jang Jun-hyuk_1 had a distance value of zero, indicating similarity. The values in the white cells represent the distance matrix values calculated using the sim2diss formula, while the values in the yellow cells are set to one, as they compare the same documents.

In the distance matrix phase, names were disambiguated based on the distance values obtained from the document similarities. The *AgglomerativeClustering* model in Python was used. When performing HAC, the number of formed clusters is uncertain. Hence, the hyperparameter value *n_clusters* determining the number of clusters was set to none. As previously described, the pre-calculated distance was used to represent the distance matrix; therefore, affinity was set to pre-computed. Additionally, distance measurement methods,

such as single, complete, and average linkages, were used. After an intrinsic evaluation, the most suitable linkage method was selected. Finally, the *distance_threshold* value, which sets the stopping criterion for the clustering process, was also determined through intrinsic performance evaluation to determine the most suitable value.

**Table 5.** Distance matrix example.

|        | Jang_0 | Jang_1 | Jang_2 | Jang_3 |
|--------|--------|--------|--------|--------|
| Jang_0 | 1      | 0      | 0.525  | 0.85   |
| Jang_1 | 0      | 1      | 0.35   | 0.8    |
| Jang_2 | 0.525  | 0.35   | 1      | 0.85   |
| Jang_3 | 0.85   | 0.8    | 0.85   | 1      |

Figure 7 shows an example of a dendrogram, using which the results of the HAC were visualized. Figure 6 shows the grouping of clusters. The red dotted line in the figure represents the stopping criterion, *distance_threshold*. Clusters grouped by the stopping criterion are marked in orange, while those not grouped are marked in blue. Clusters divided by the stopping criterion indicate individuals with the same name.

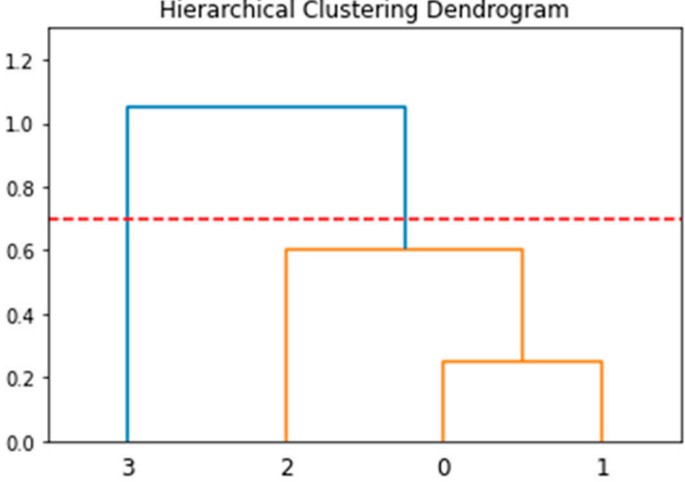

**Figure 7.** Example of a dendrogram.

As an example from Table 5, when setting the stopping criterion to 0.4, the documents "Jang_0", "Jang_1", and "Jang_2" were grouped into one cluster, while "Jang_3" was represented in a different cluster. In other words, two authors were distinguished.

### 3.6. Multi-Classifier

In this study, name disambiguation was conducted by collecting data in real time from academic search websites. Metadata provided by heterogeneous academic search sites vary depending on the site characteristics. Therefore, name disambiguation needs to consider these varying characteristics. Not all academic search sites hold the metadata required for name disambiguation. Hence, conducting name disambiguation using the available metadata from these academic search services was necessary.

The name disambiguation method proposed in this paper employed both rule-based and deep learning schemes. The rule-based approach has the advantage of quickly disambiguating names when documents with the applicable metadata are input based on set rules. Conversely, although the deep learning scheme requires training time, it can perform name disambiguation even in the absence of essential metadata. Therefore, even with the same metadata, diverse name disambiguation schemes can be applied to obtain results. In this paper, a multi-classifier that can select the appropriate name disambiguation scheme using the collected metadata is proposed. Using the multi-classifier, even if missing data

are input, the appropriate name disambiguation scheme can be chosen to derive results. This method is scalable, i.e., it can allow future addition of new academic search services or new name disambiguation classifiers based on the results from the multi-classifier.

Figure 8 displays the schematic of a multi-classifier that considers metadata from various academic search services and accordingly selects the appropriate name disambiguation method. When embedded name disambiguation data from name disambiguation candidates are input, the proposed multi-classifier selects between the presented rule-based and learning name disambiguation, considering the presence or absence of input metadata attributes.

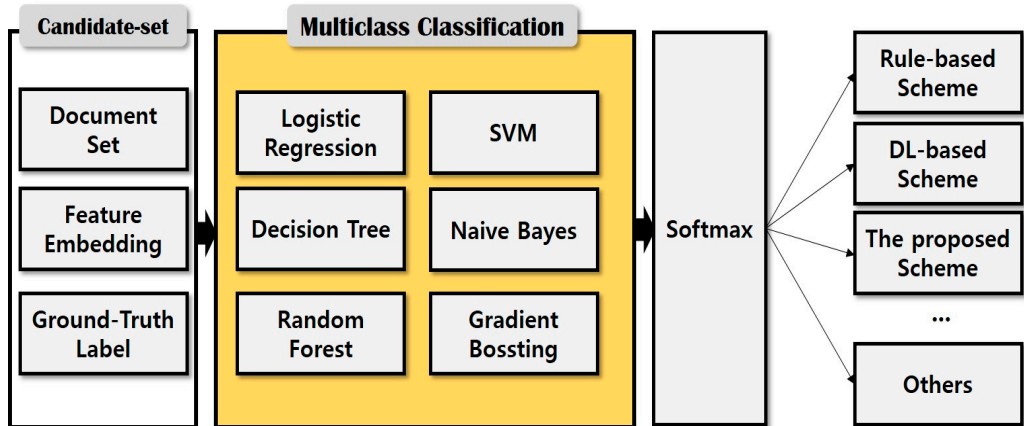

**Figure 8.** Overview of the multi-classifier structure.

The multi-classifier operated through the following steps. First, we perform the feature embedding. The attributes of candidate documents for name disambiguation are transformed into vector representations via a feature embedding process, such as word2vec, doc2vec, and FastText. Next, we conduct performance evaluations for each name disambiguation schemes by using the transformed feature value. Finally, by comparing the performances, we select the best scheme as the ground truth label for the transformed feature value. We employ the labels to train the proposed multi-classifier. After all the ground truth labels were generated, training was conducted using these data and various existing multi-classifiers. The trained multi-classifier then received input values that were converted to vectors through the feature embedding process of site-specific metadata. Ultimately, the multi-classifier produced the most appropriate identification scheme as its output. Thus, selecting the most suitable name disambiguation scheme in real-world environments becomes possible where various metadata are generated. Furthermore, even when a new identification scheme is introduced, the multi-classifier can be extended by adding one label, enabling the utilization of an expandable multi-classifier model.

## 4. Performance Evaluation

### 4.1. Performance Evaluation Environment

The performance of the proposed name disambiguation scheme was evaluated based on the termination criterion and linkage method of the HAC to validate its utility. In this study, the performance of the proposed and existing rule- and deep learning-based name disambiguation schemes was comparatively evaluated [7,13]. Moreover, the performance of the multi-classifier, which selects a name disambiguation scheme based on attributes, was evaluated. The performance evaluation environment is summarized as follows. The performance evaluation was conducted on a system built with an Intel(R) Core(TM) i5-9600K CPU @ 3.70 GHz 64-bit processor, Santa Clara, CA, USA, and 32 GB of memory. The proposed scheme was implemented using Python 3.8.12 in the Python Anaconda environment, and machine-learning libraries, such as sklearn, keras, tensorflow, and the matplotlib library for data visualization, were used. The collected dataset is sum-

marized in Table 6. The dataset used for the performance evaluation includes all research results published in the last 1–10 years. It includes a total of 23,563 entries from academic and project databases, such as NTIS, DBPIA, KCI, and SCIENCEON. These entries are based on search keywords, like "database indexing", "IoT applications", "cloud computing", "big data social network", "AI verification", "virtual reality", and "steering control". Among all the collected research materials, 2460 name disambiguation candidate groups were created, targeting materials with the same author name appearing in two or more research outputs. The attributes of the collected data consisted of paper ID, co-authors, author ID, document title, academic journals and conferences, affiliation, publication year, etc. The performance evaluation of the proposed name disambiguation scheme consisted of its own performance evaluation, comparative performance evaluation with the existing name disambiguation schemes, and performance evaluation of the multi-classifier. To measure the accuracy of the proposed scheme, the precision, recall, and F1-measure were calculated. An intrinsic performance evaluation of the proposed scheme and a comparative evaluation with other name disambiguation schemes were conducted.

**Table 6.** Dataset.

| Keyword | Period | NTIS | SCIENCEON | DBPIA | KCI | Total |
|---|---|---|---|---|---|---|
| Database and Index | 10 | 208 | 47 | 966 | 76 | 1297 |
| IoT and Application | 5 | 2889 | 138 | 3302 | 261 | 6590 |
| Cloud Computing | 3 | 981 | 138 | 1459 | 293 | 2871 |
| Bigdata and SNS | 10 | 471 | 153 | 1910 | 71 | 2605 |
| AI and Verification | 2 | 2826 | 104 | 2176 | 227 | 5333 |
| AR/VR | 1 | 540 | 85 | 1870 | 335 | 2830 |
| Steering and Control | 10 | 289 | 76 | 1561 | 111 | 2037 |
| Total | - | 8204 | 741 | 13,244 | 1374 | 23,563 |

*4.2. Intrinsic Performance Evaluation*

Clustering in HAC requires setting a termination criterion to distinguish unique clusters. In this section, the performance evaluation based on the termination criterion of HAC is discussed. In the proposed method, authors may be clustered differently depending on the termination criterion when using HAC. Therefore, in this study, various termination criterion values were set, and experimental evaluations were conducted as an intrinsic performance evaluation method.

Figure 9 displays the performance evaluation results based on the termination criterion. The results for the precision and F1-measure are represented in a bar graph. Performance evaluation was conducted by changing the termination criterion values from 0.2 to 0.6. The termination criterion of 0.2, which showed the highest F1-measure value of 0.95, was determined to be the most suitable termination criterion. A termination criterion of 0.2 means that if the distance between documents (or clusters) is closer than 0.2, they are clustered, and if not, they are not clustered. Through experimental evaluations, the termination criterion of 0.2 was used as the HAC standard in the proposed scheme.

A performance evaluation was conducted based on the linkage method setting, which is one of the hyper-parameters of HAC. In the proposed scheme, when implementing the rule-based name disambiguation method using HAC, the clustering results can vary depending on the linkage method, similarly to that with the termination criterion. Therefore, in this study, an experimental evaluation for each linkage method was conducted as an intrinsic performance evaluation method to determine the optimal linkage method. The termination criterion was set to 0.2, as determined through performance evaluation in the previous step. Figure 10 displays the performance evaluation results based on the linkage method. Three linkage methods were compared: single, complete, and average, excluding Ward's linkage, which cannot be used in HAC. According to the experimental results, the complete linkage method exhibited a higher F1-measure value than the other two methods.

Therefore, the complete linkage method, which yielded the highest scores for precision and F1-measure, was adopted as the linkage method in the proposed scheme.

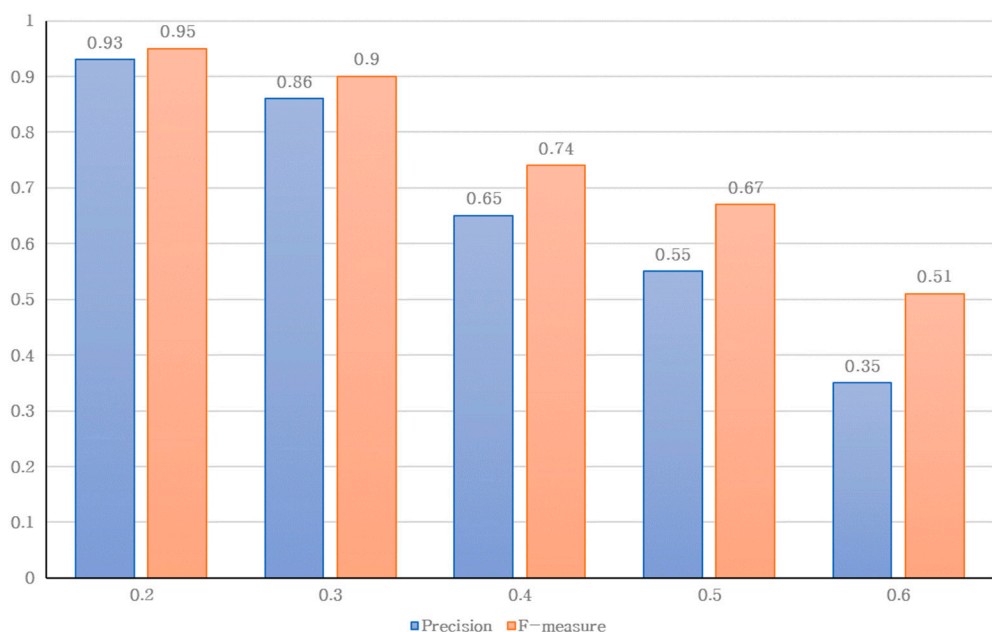

**Figure 9.** Precision and F1-measure according to the HAC termination criterion.

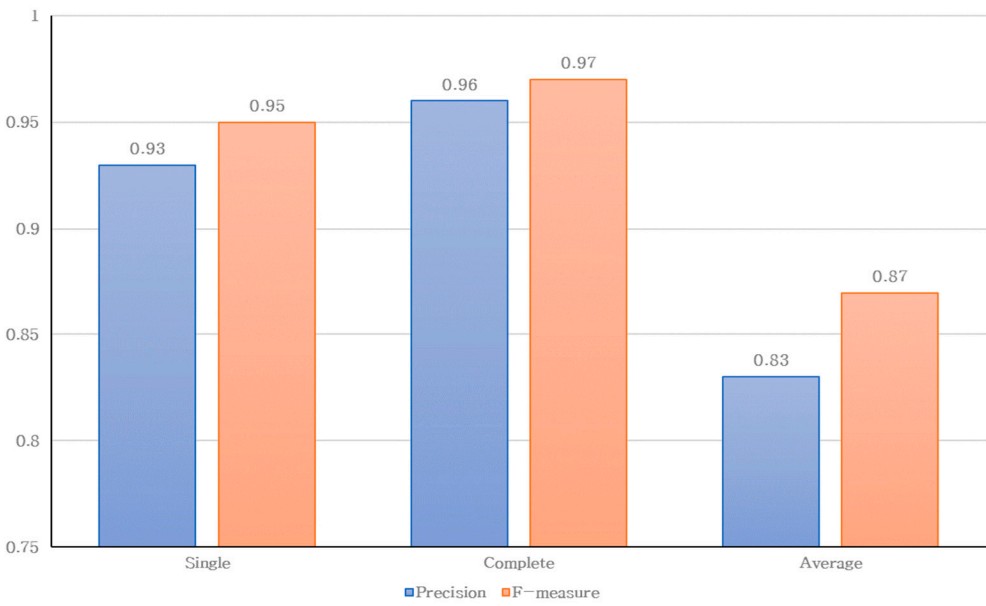

**Figure 10.** Precision and F1-measure according to the HAC linkage method.

### 4.3. Comparative Performance Evaluation of Name Disambiguation Schemes

A performance comparison with existing name disambiguation schemes was conducted to demonstrate the superiority of the proposed method. In this study, two schemes were compared: (1) the Protasiewicz method [7], an existing rule-based name disambiguation scheme, that uses the attributes of papers to create rules and runs HAC with the weights of these rules to disambiguate names. (2) The Chen Ya method [13], an existing deep learning-based name disambiguation scheme, that learns paper attributes using a GCN and then runs HAC with the resulting weights to disambiguate names. The HAC of the proposed scheme was set with a termination criterion of 0.2 and used the complete linkage method. The dataset used for performance evaluation is summarized in Table 6.

The superiority of the proposed method was demonstrated through a comparative performance evaluation of precision, recall, and F1-measure for name disambiguation accuracy between the proposed and existing methods.

Figure 11 shows the results of the comparative performance evaluation of precision based on the name disambiguation schemes. The precision of the proposed scheme exhibited a very high performance, with scores >0.99 for all keywords. However, the existing rule-based scheme, the Protasiewicz method, showed a decent average performance but underperformed for certain keywords. Additionally, the deep learning-based scheme demonstrated the poorest average performance. The data used for the performance evaluation were collected from actual academic search services. This indicated that existing studies need more detailed preprocessing and analysis when performing name disambiguation analysis based on real data. Furthermore, it implies that these characteristics must be reflected, since not all academic search services provide the same metadata.

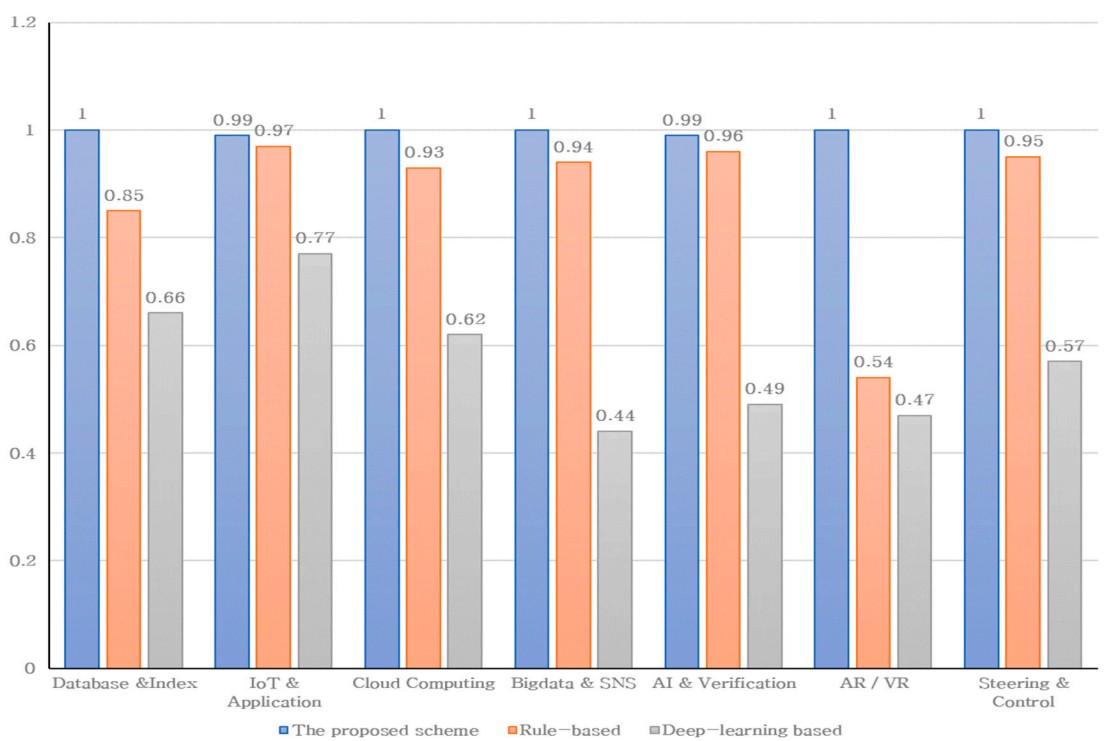

**Figure 11.** Precision according to the name disambiguation schemes.

Figure 12 displays the results of the comparative performance evaluation of recall based on the name disambiguation schemes. The recall of the proposed scheme demonstrated a very high performance, with scores >0.97 for all keywords. The existing deep learning-based Chen Ya method showed excellent performance for keywords "cloud computing", "AI verification", and "steering control"; however, the proposed method outperformed for all keywords.

Figure 13 displays the results of the comparative performance evaluation of F1-measure based on the name disambiguation schemes. The F1-measure of the proposed scheme demonstrated a very high performance, with scores >0.98 for all keywords. Compared with the existing rule-based Protasiewicz scheme and deep learning-based Chen Ya scheme, the proposed scheme exhibited higher performance across all keywords, thereby proving its superiority. A deep learning-based scheme is useful for diversifying keywords or feature dimensions. However, unlike the rule-based scheme, it is difficult to determine the similarities of affiliation or co-author name. For this reason, the deep learning–based scheme shows lower performance than the proposed rule-based scheme.

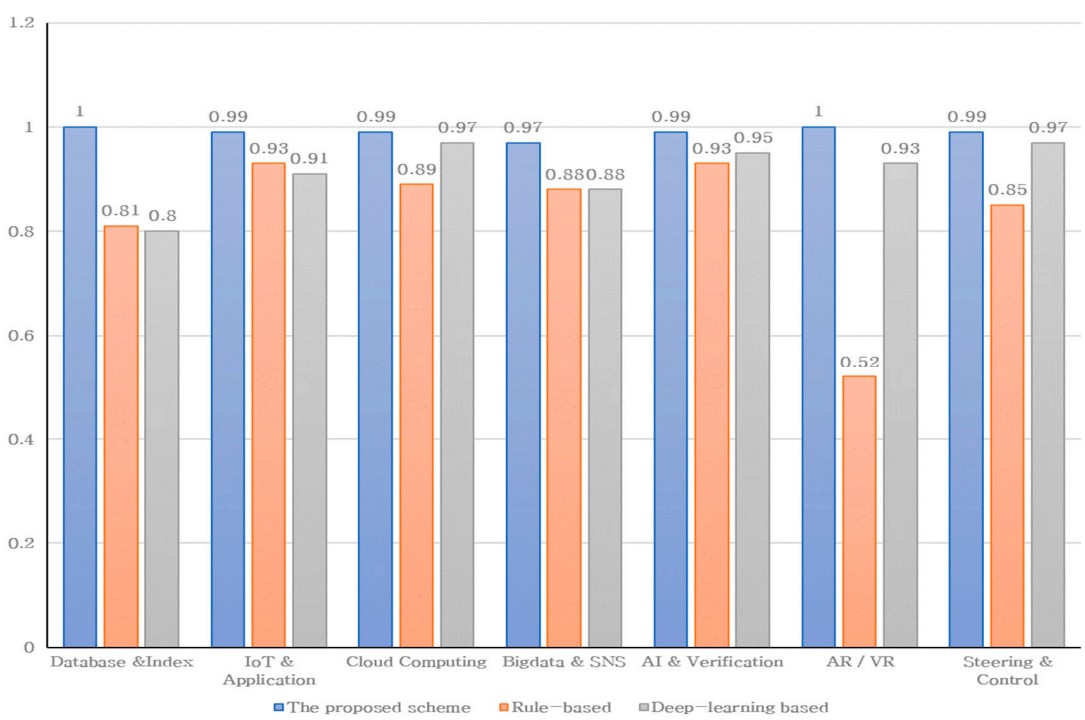

**Figure 12.** Recall based on the name disambiguation schemes.

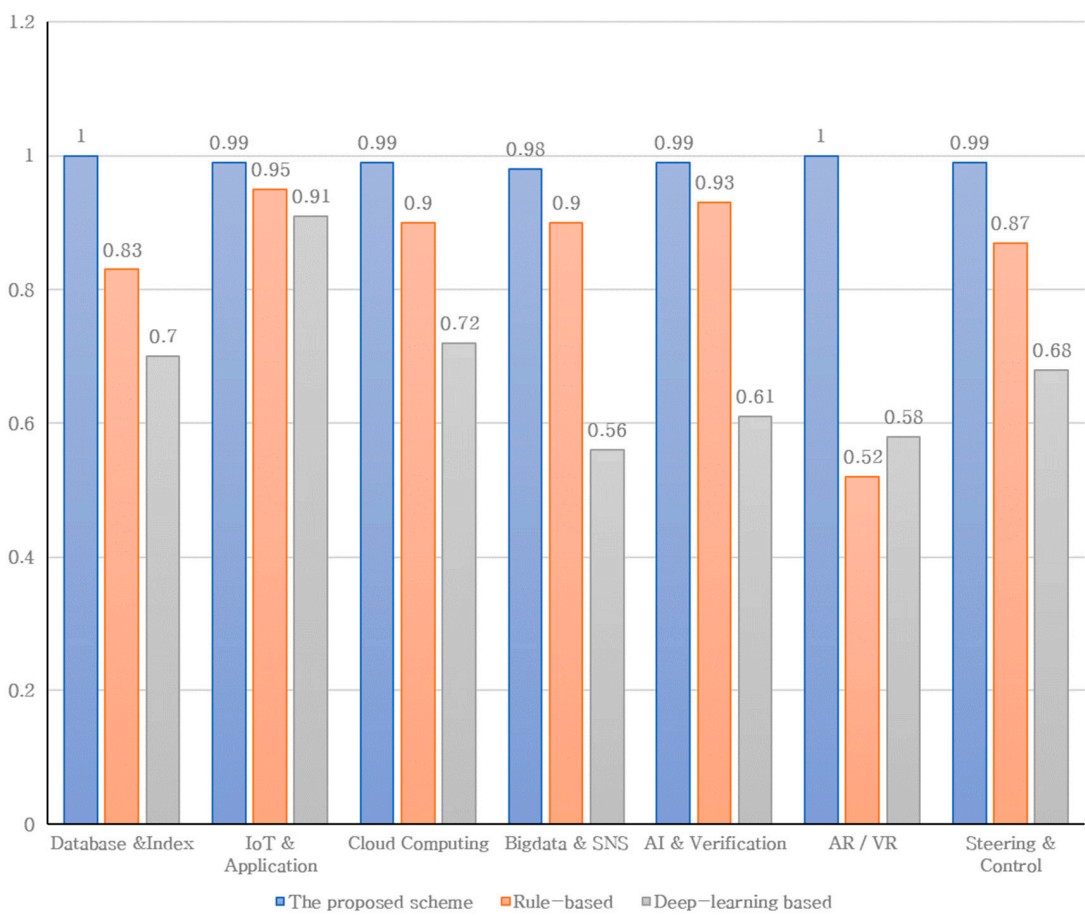

**Figure 13.** F1-measure based on the name disambiguation schemes.

### 4.4. Multi-Classifier Performance Evaluation

In the performance evaluation of the multi-classifier, the proposed scheme selected either the rule-based or deep learning discrimination method based on attributes using machine learning. The machine-learning models used in the multi-classifier include a total of four classification schemes: support vector classification (SVC), linear SVC, random forest, and naive Bayes. Through the multi-classifier performance evaluation, the most appropriate multi-classifier scheme was determined. To measure the accuracy of the proposed method, performance was evaluated by calculating precision, recall, and F1-measure. The performance evaluations of the classifiers were conducted as follows. First, we test the optimal hyperparameter for each classifier based on exactly the same training set. The training set was designated as 80% of the dataset. After that, the performance evaluation for each keyword was conducted based on the optimized model. For example, the hyperparameters of the SVC model are as follows: C, gamma, and the kernel are derived to be 0.1, 0.001, and rbf, respectively. The alpha value of the naïve Bayse model is set to 1.0.

For the input of the multi-classifier, two methods were compared: one that represents attributes in a binary form (1 and 0) and another that embeds attributes into vector values using word2vec(W2V) for each attribute. The first method transforms values based on the presence or absence of an attribute. If the attribute is present, it is represented as one, and if absent, it is represented as zero for classifier training. Figure 14 shows the results of the multi-classifier performance evaluation based on binary attribute embedding. All four classification schemes displayed similar precision; however, the F1-measure of SVC and random forest showed values >0.7, indicating approximately 8% higher performance than linear SVC and naive Bayes schemes.

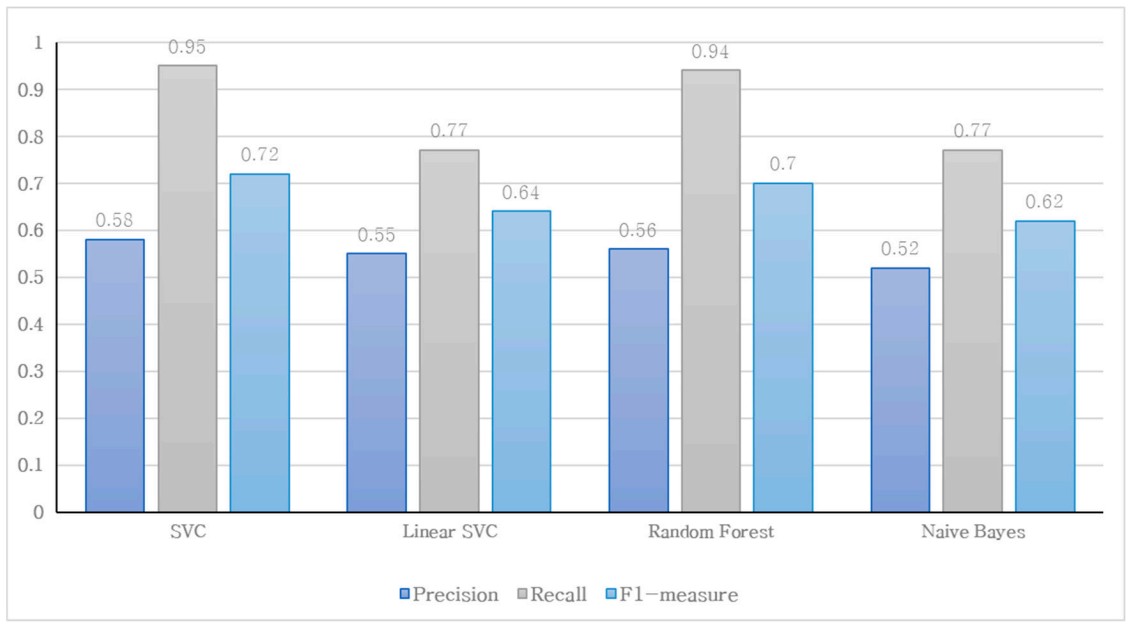

**Figure 14.** Comparison of multi-classifier performance based on binary attribute embedding.

The second method involves evaluating the performance of a multi-classifier based on W2V, one of the most renowned feature embedding schemes. Figure 15 shows the results of the multi-classifier performance evaluation based on W2V. The performance evaluation results showed that the random forest scheme exhibited outstanding precision and an F1-measure value of 0.98. This was 28% higher than the results of the random forest scheme embedded using 1 s and 0 s. Additionally, SVC also displayed an F1-measure value of 0.98, which was 23% better than that of the previous method. In conclusion, for the multi-classifier, use of values transformed through W2V for training is more suitable rather than relying solely on the presence or absence of attributes.

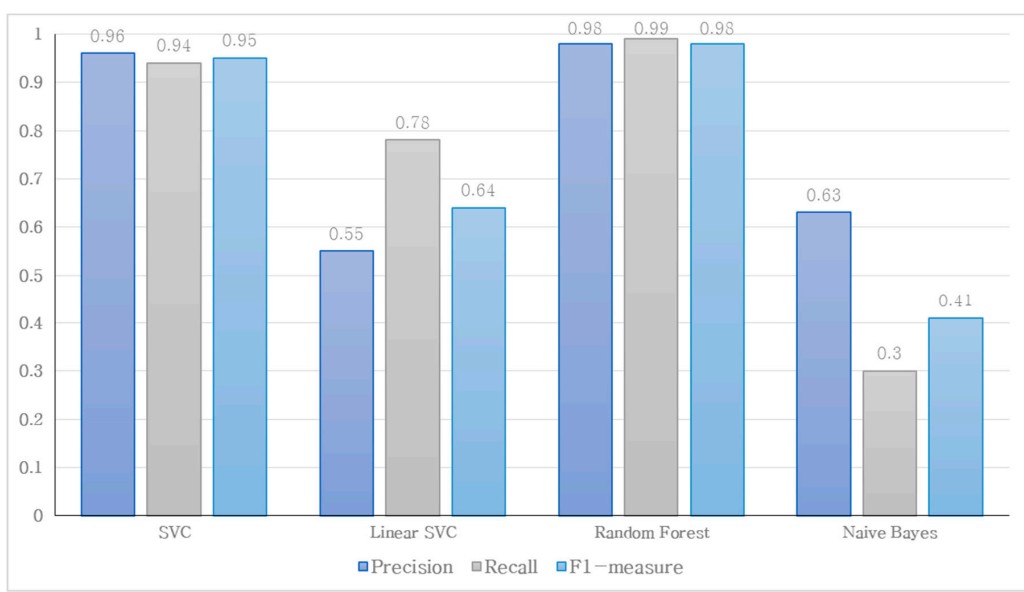

**Figure 15.** Comparison of multi-classifier performance based on W2V.

## 5. Conclusions

In this paper, we proposed a name disambiguation scheme based on heterogeneous academic search sites. The proposed scheme integrated and collected research outcomes provided by heterogeneous academic search sites. Using the collected data, name disambiguation was performed using clustering schemes based on necessary attributes. Moreover, the proposed method was compared with and evaluated against traditional rule-based name disambiguation schemes and deep learning-based name disambiguation schemes. Considering the metadata provided by academic search sites, we proposed a multi-classifier capable of selecting a more accurate name disambiguation scheme. The proposed multi-classifier selects a more precise name disambiguation scheme. The performance evaluation of the proposed method showed an exceptionally high F1-measure value of 0.99, confirming its suitability as the most apt scheme for name disambiguation. In performance evaluation of the proposed multi-classifier, we showed that very high performance was obtained by using W2V. We verify the excellence of the proposed scheme through various performance evaluations. The proposed name disambiguation scheme considers various academic search sites. In addition, the proposed scheme proved that expandability through a multi-classifier is possible. In future, we plan to expand the proposed method to a multi-language-based name disambiguation scheme. In addition, we will conduct research on performance improvement through the combination of rule-based and GCN-based name disambiguation scheme.

**Author Contributions:** Conceptualization, D.C., J.J., S.S., H.L., J.L., K.B. and J.Y.; methodology, D.C., J.J., S.S., H.L., J.L., K.B. and J.Y.; validation, D.C., J.J., H.L., J.L. and K.B.; formal analysis, D.C., J.J., S.S., H.L., J.L. and K.B.; writing—original draft preparation, D.C., J.J., S.S. and K.B.; writing—review and editing, J.Y. All authors have read and agreed to the published version of the manuscript.

**Funding:** This work was supported by the National Research Foundation of Korea (NRF) grant funded by the Korean government (MSIT) (No. 2022R1A2B5B02002456, No. RS-2022-00166906), the Korea Association of University, Research Institute and Industry (AURI) grant funded by the Korean Government (Ministry of SMEs and Startups; MSS) (No. S3047889, HRD program for 2021), and by "the Ministry of Science and ICT (MSIT), Korea, under the Grand Information Technology Research Center support program (IITP-2023-2020-0-01462) supervised by the Institute for Information and Communications Technology Planning and Evaluation (IITP).

**Institutional Review Board Statement:** Not applicable.

**Informed Consent Statement:** Not applicable.

**Data Availability Statement:** The data presented in this study are available on request from the corresponding author. The data are not publicly available due to data sets provided by the company.

**Conflicts of Interest:** The authors declare no conflicts of interest.

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
