# Peer review of "Name Disambiguation Scheme Based on Heterogeneous Academic Sites"

_applsci, doi:10.3390/app14010192_

Round 1

Reviewer 1 Report

Comments and Suggestions for Authors

Line 53. How about surnames? Avoid confusion. 

Line 80. Indicate what classifier this is.

Line 88. Use better-fitting instead of superiority 

Line 94. Is the weight stable or dynamic? Indicate (a coefficient assigned to a number in a computation (if not dynamic))

Line 104. Convince readers why HAC is suitable to the methodology.  As known, HAC merges the most similar clusters together, starting with each data point as a separate cluster.

For Sec. 2.1. Prepare a flow chart(s) that shows the steps in a process mentioned in this section. This will be helpful in visualising the sequence of actions and information needed for training, documenting, planning, and decision-making associated with the methodology, and also indicates the differences between yours and the existing ones.  

Line 168. Give examples of academic search services (as Scopus?)

Line 201. ‘both the rule-based and deep-learning schemes’ … , Is this done step by step or all together? Give further detail about the Author’ name disambiguation, as it is not clear. 

Line 202. Give more detail about the distance matrix that is used for the methodology. (what program (or in-house), is it Ultrametric? et cetera)

For Sec. 3.2.  Explain the algorithm that is used for metadata to distinguish journals, academic journals, and proceedings. Curious about how well the algorithm works and whether any validation tests were done before use?

Figure 2 is not clear. Check its resolution. 

Line 350. Give the definition of ‘m’ in the affiliation formula.  

Line 351. 0.83? explain this. 

Line 357. Explain why the weight of the co-author ratio is low.  Any drawback affecting the overall outcome?

Figure 4. Draw the figure better.  In some cases, the positive examples will be closer to the anchor, while the negative examples will be farther from it after the training. 

Line 503.  The multiclass classification in Figure 6 needs to be explained better in the paragraph.

Line 528. Express the computing time for the methodology. 

Author Response

Dear reviewer,

We would like to sincerely thank you for your attentive indications and good comments. We did our best and partially rewrote our manuscript in order to reflect your comments. Please refer to the attached file about the detailed revision.

Many thanks.

Jaesoo Yoo

Reviewer 2 Report

Comments and Suggestions for Authors

The manuscript introduces a name disambiguation scheme to address challenges in identifying researchers with the same name across heterogeneous academic sites. The proposed scheme collects and integrates research results from diverse sources, emphasizing disambiguation-relevant attributes. Utilizing clustering techniques, it identifies individuals with identical names. The study compares a rule-based algorithm with an existing deep learning-based method, employing a multiclass classification approach for accurate disambiguation. Performance evaluations, including accuracy, recall, and F1-measure, demonstrate the superior effectiveness of the proposed scheme in name disambiguation. Below are my some concern:

• The primary motivation for the research work, along with its significance, should be incorporated into the introduction section or a relevant segment.

• The benefits of the proposed work need to be elucidated in the conclusion section.

• The author is advised to engage a language expert to scrutinize the paper for grammatical issues and typos.

• The conclusion is deemed unsatisfactory; it should be succinct, clearly stating the contributions made.

• The inclusion of future directions in the conclusion section is recommended.

Comments on the Quality of English Language

Moderate editing of English language required

Author Response

(The authors gave the same response as above.)

Reviewer 3 Report

Comments and Suggestions for Authors

The paper presents a name disambiguation scheme based on herterogeneours academic search sites. The authors focused on critical feature extraction from the data and then applied clustering method to differentiate persons with the same name. The proposed scheme was evaluted with regard to different metrics (e.g., precison, F1 score, etc.) and the performance was compared with existing algorithms, showing that the proposed scheme was effective and competitive. Overall, the paper is well organized. The research topic is interesting and important, which may lead to practical applications. 

I think the topic is original and relevant in the field. Although the research problem has been studied by many scholars and the methods the authors adopted, rule-based learning and deep learning, were already applied to the field by other researchers, the authors proposed an augmented scheme which can better extract crucial features (e.g., standardizing researchers' affiliations) for name disambiguation from the data collected through heterogeneous academic search sites. Plus, they designed a multiclassification method which can choose the most suitable scheme for name disambiguation. 

The proposed scheme is effective regarding its accuracy and outperforms some existing methods, which may lead to practical applications.   

Minor wording problems exist throughout the paper which can be fixed by proofreading (a few examples shown below). Also, the dimensions of Figures 8 to 14 are unnecessarily large. I suggest the authors consider reducing each figure to about 60% or 70% of its current size. 

A few examples of wording issues: 

(1) lines 20-21 "Additionally, the proposed rule-based algorithm name disambiguation method and the existing deep learning-based identification method", the sentence misses a verb. 

(2) line 148, "if metadata not considered in the name disambiguation dataset are absent", missing "the" in front of "metadata". Also, the word "not" should be removed if I did not misunderstand the meaning. 

(3) line 159, "Additionally, as academic search services provide research materials specific to their purpose", "purpose" should be in plural format. I suggest changing "their purpose" to "their own purposes". 

(4) through the paper, "metadata" is used as a plural noun. However, "data" is more frequently used with a singular verb in data science.   

The authors may consider adding more references of the latest publications. Also, references [22]-[25] are common Python modules used for research, no need to cite them in scientific writing or include them in references.   

The dimensions of Figures 8 to 14 are unnecessarily large. Please consider reducing each one to about 60% or 70% of its current size.

Comments on the Quality of English Language

Minor writing issues exist throughout the paper. For example, in lines 20-21 "Additionally, the proposed rule-based algorithm name disambiguation method and the existing deep learning-based identification method", the sentence has grammatical error (missing verb). Proofreading is required before the paper is accepted. 

Author Response

(The authors gave the same response as above.)

Reviewer 4 Report

Comments and Suggestions for Authors

## Review summary

Quality: The quality of the presented approach seems adequate, although not extraordinary. The approaches and techniques used are fairly standard but are combined in an innovative manner. A more complex experimental design of a performance evaluation step (Section 4) could, in my opinion, provide more realistic insights. In particular, I strongly suggest including some sort of cross-validation approach and accessing standard errors in performance metrics.

Clarity: The paper is written clearly. The introduction provides clear motivation, although some definitions should be elaborated (e.g., what exactly is name diasambiguation?). The method descriptions could benefit from more details on the actual data collection process (clearly describe all data sources you used with a complete list of attributes you employed in the study).

Originality and Significance: The problem of name disambiguation has been widely studied, but the presented analysis is unique and presents a certain degree of novelty. Overall, the work provides an incremental application of existing methods and advances.

## Major issues

1. The related work must be expanded to present the reader with a general overview of the field; in addition, some important references are missing [e.g., 1‒4].

2. The experimental phase should be elaborated, at least to include some sort of cross-validation to assess variability among classifiers.

3. The deep learning approach has demonstrated the lowest performance; please elaborate on those differences, demonstrate if differences between classifiers were statistically significant, and explain potential reasons for variation.

## Minor issues

1. Page 1, Abstract: Add a brief overview of the main findings to the abstract, and note to the reader what your paper adds to the body of science.

2. Page 7, Figure 2: Replace Figure with a table; the current figure is ugly and hard to read.

3. Page 11, line 411: Probably a typo.

4. Page 15, table 5: Replace the table with an inline description.

## Additional references

[1] Ferreira, A. A., Gonçalves, M. A., & Laender, A. H. (2012). A brief survey of automatic methods for author name disambiguation. Acm Sigmod Record, 41(2), 15-26.

[2] Hussain, I., & Asghar, S. (2017). A survey of author name disambiguation techniques: 2010–2016. The Knowledge Engineering Review, 32, e22.

[3] Milojević, S. (2013). Accuracy of simple, initials-based methods for author name disambiguation. Journal of Informetrics, 7(4), 767-773.

[4] Sanyal, D. K., Bhowmick, P. K., & Das, P. P. (2021). A review of author name disambiguation techniques for the PubMed bibliographic database. Journal of Information Science, 47(2), 227-254.

Comments on the Quality of English Language

N/A.

Author Response

(The authors gave the same response as above.)

Round 2

Reviewer 2 Report

Comments and Suggestions for Authors

None